# Surgical Treatment of Spinal Meningiomas in the Elderly (≥75 Years): Which Factors Affect the Neurological Outcome? An International Multicentric Study of 72 Cases

**DOI:** 10.3390/cancers14194790

**Published:** 2022-09-30

**Authors:** Gabriele Capo, Alessandro Moiraghi, Valentina Baro, Nadim Tahhan, Alberto Delaidelli, Andrea Saladino, Luca Paun, Francesco DiMeco, Luca Denaro, Torstein Ragnar Meling, Enrico Tessitore, Cédric Yves Barrey

**Affiliations:** 1Department of Spine and Spinal Cord Surgery, Hôpital Pierre Wertheimer, Hospices Civils de Lyon, Claude Bernard University of Lyon 1, 59 Boulevard Pinel, 69677 Lyon-Bron, France; 2Department of Neurosurgery, Geneva University Hospitals, Faculty of Medicine, University of Geneva, 1205 Geneva, Switzerland; 3Department of Neurosurgery, GHU Paris—Psychiatrie et Neurosciences, Sainte-Anne Hospital, 75014 Paris, France; 4Institut de Psychiatrie et Neurosciences de Paris (IPNP), UMR S1266, INSERM, IMA-BRAIN, 75014 Paris, France; 5Academic Neurosurgery, Department of Neurosciences, University of Padova, 35128 Padova, Italy; 6Department of Molecular Oncology, British Columbia Cancer Research Centre, Vancouver, BC V5Z 1L3, Canada; 7Department of Pathology and Laboratory Medicine, University of British Columbia, Vancouver, BC V6T 1Z7, Canada; 8Neurosurgery Department, Fondazione IRCCS Istituto Neurologico Nazionale “C. Besta”, 20133 Milan, Italy; 9Department of Oncology and Hemato-Oncology, University of Milan, 20122 Milan, Italy; 10Department of Neurological Surgery, Johns Hopkins Medical School, Baltimore, MD 21205, USA; 11Laboratory of Biomechanics, ENSAM, Arts et Metiers ParisTech, 153 Boulevard de l’Hôpital, 75013 Paris, France

**Keywords:** spinal meningiomas, spinal cord, neurosurgery, elderly, functional outcome, geriatric surgery, quality of life, functional disability

## Abstract

**Simple Summary:**

Spinal meningiomas (SMs) are slow growing lesions, often occurring in middle- and old-aged patients. Few data about age-related prognostic factors are available in the literature to date. We analyzed a series of elderly patients undergoing surgery for a SM in the last twenty years in four different European tertiary referral centers. This work aimed to assess the surgical outcome and to identify possible outcome predictors. In this international multicentric retrospective study involving 72 patients older than ≥75 years, we highlight that functional preoperative score (according to modified McCormick scale) and age at surgery correlate with functional outcome.

**Abstract:**

(1) Background: With the increasing life expectancy in the Western world, an increasing number of old patients presents with spinal meningioma. Considering the benign nature of these tumors, the functional outcome remains of great importance, since more people reach old age in general conditions of well-being and satisfactory autonomy. (2) Methods: We conducted an international multicenter retrospective study to investigate demographic, clinical and radiological data in a population of elderly patients (≥75 years of age) undergoing surgery for SM from January 2000 to December 2020 in four European referral centers. The aim was to identify prognostic and predictive factors for a good postoperative functional outcome. (3) Results: 72 patients were included in the study. Complete tumor resection (Simpson I or II) was achieved in 67 (95.7%) cases. Intraoperative complications were reported in 7 (9.9%) patients while postoperative complications were found in 12 (16.7%). An excellent general postoperative status (McCormick I and II) was achieved in 65.3%. Overall, surgical resection had a good impact on patients’ functional outcome (86.1% either showing an improvement or maintaining a good preoperative status). Uni- and multivariate analyses found that both age and preoperative modified McCormick independently correlated with relative outcome (coeff = −0.058, *p* = 0.0251; coeff = 0.597, *p* < 0.0001) and with postoperative status (coeff = 0.058, *p* = 0.02507; coeff = 0.402, *p* = 0.00027), respectively. (4) Conclusions: Age and preoperative modified McCormick were found to be independent prognostic factors. Nevertheless, advanced age (≥75), per se, did not seem to contraindicate surgery, even in those with severe preoperative neurological deficits. The functional results sustain the need for surgical resection of SM in the elderly.

## 1. Introduction

Spinal meningiomas (SMs) predominantly occur in middle and old-aged patients, mostly female [1,2,3,4,5]. Surgery is the treatment of choice, generally associated with a good neurological outcome notwithstanding the advanced age [6,7,8,9,10,11,12]. With increasing life expectancy in high-income countries, the elderly population is expected to grow further in the future [13,14]. Nonetheless, the interest in the literature for this age group is expanding, evidenced by an exponential increase of papers published about geriatric surgery in the last decade [15]. Moreover, long-term disability in older patients is associated with low life expectancy [16], and surgery should therefore aim to maintain life span, dignity, and an appropriate quality of life [17]. Since SMs are benign and slow-growing lesions, the diagnosis is often delayed and in the elderly the symptoms may be attributed to pre-existing age-related pathologies, rather than to a spinal tumor [11]. Furthermore, older patients are prone to present at diagnosis with motor impairment and bladder disturbances [11], to be in a dependent state [8,9,10], or to be in rehabilitation before surgery [11]. The available literature provides few descriptions of elderly patients surgically treated for SMs, mostly included in large series of younger subjects [1,6,7,18,19,20,21,22,23,24,25]. Only two series specifically focused on older patients [9,10] while in anotherr two papers the authors compared the group of elderly with the younger patients [8,11,26]. Besides, there is no consensus on the cut-off age (>70 years or >75 years old) to consider the patient as elderly in the above-mentioned studies. Despite the increasing incidence of SMs in elderly and the rate of positive functional outcome after surgery, there is still reluctance to propose surgery to these patients. This attitude might be due to an expected higher risk of systemic postoperative complications [11,12,27], in consequence of peculiar physiology and age-related changes in organs [17].

The aim of this study was to perform a thorough analysis of demographic, radiological, and clinical findings evaluating their influence on outcome in an international multicentric cohort of elderly patient (≥75 years of age) surgically treated for a SM over a period of 20 years.

## 2. Materials and Methods

### 2.1. Study Design and Patient’s Selection

An international multicentric retrospective cohort study was conducted from the database of 4 European tertiary referral Neurosurgical Departments (Hospices Civil de Lyon, France; Hôpitaux Universitaires de Genève, Switzerland; Azienda Ospedale-Università Padova, Italy; and Instituto Neurologico C. Besta of Milano, Italy). Patients who underwent surgical resection for a SM from 1 January 2000 to 31 December 2020 were screened.

Inclusion criteria were (1) age ≥75 years and (2) surgery for resection of a SM between 2000 and 2020. The only exclusion criterion was (1) no clinical follow-up.

All the patients were operated by posterior microsurgical approach, and the resection was classified according to the Simpson grading system [28].

The manuscript was written according to the Strengthening the Reporting of Observational Studies in Epidemiology checklist.

Ethical approval was obtained by an institutional review board (Lyon N°21_5486).

### 2.2. Variables and Data Sources

The electronic medical records and MR images were reviewed to obtain the following data:

Preoperative data: demographics, body mass index (BMI), major comorbidities, (i.e., ischemic heart disease, peripheral artery disease, chronic obstructive pulmonary disease, active cancer, diabetes mellitus, severe renal insufficiency with dialysis, and neurodegenerative disease), presence of NF2, American Society of Anesthesiologists Classification (ASA score), duration of symptoms, preoperative motor function (MRC), preoperative functional status according to the modified McCormick scale (mod McCormick Table 1), walking distance, and sphincter dysfunction.

Radiological features: localization and number of lesions, number of involved levels, dural attachment and the presence of dural tails, T2-weighted images changes of the spinal cord. These features were obtained by reviewing preoperative MRI. Tumor volume (cm^3^), which was manually segmented using Brainlab Elements^®^ on axial T1-weighted images with contrast medium (slice thickness from 3 to 4 mm).

Surgical and pathological data: use of intraoperative neuromonitoring, Simpson grading of resection, intraoperative complications (i.e., nerve root or cord lesion, hemorrhage). Postoperative complications were classified as neurological (i.e., hematoma with symptoms of root and/or cord compression, CSF leak and new onset neurological deficits), surgical infection or wound-related, and general complications (i.e., bedrest-related complications); surgery for complications, WHO grading, and histological type.

Outcome measures: Clinical outcome was evaluated with mod McCormick scale at last follow-up. Patients with a mod McCormick I and II at follow-up were considered having a good overall status; mod McCormick III–V were considered as poor overall status. We calculated the relative outcome (Δ mod McCormick) as the difference between preoperative and follow up mod McCormick value. A positive Δ mod McCormick was considered to select the patients who benefit most from surgical treatment (good relative outcome). When available, follow-up MRIs were reviewed to assess the presence of T2-weighted image changes of the spinal cord.

### 2.3. Statistical Analysis

For univariate analyses, associations between variables and outcome groups were calculated with Fisher’s exact test for categorical and semi categorical variables. Continuous variable differences between groups were computed with Wilcoxon signed-rank test. Missing data were considered as part of the analyses but not calculated as percentages in the text and in the tables. For multivariate analyses, co-variates that resulted significantly associated with outcome in the univariate analyses were fitted to a linear model including the outcome measure as a dependent variable. Subsequently, analysis of variance for each of the models was computed and reported. All statistical analyses were performed in R 4.1.1.

## 3. Results

### 3.1. Preoperative Data

A total of 245 patients were identified, of whom 72 were selected according to the inclusion and exclusion criteria. Median age of 78.5 (Q1–Q3 range, 77.0–84.0), 61 (84.7%) were female and the median follow-up was 12 ± 16.6 months. Mean BMI score was 31.4 ± 17.9. At initial assessment, 44 (61.1%) patients presented without previous major comorbidities. One (1.4%) patient was affected by type II neurofibromatosis, and five (6.9%) patients had a previous surgery for spinal meningioma. The ASA score was ≥3 in 19 (41.3%) cases. The mean duration of symptoms before surgery was 12 months (Q1–Q3 range, 6.8–17.0), with 72.2% of the patients being symptomatic for 9 months or more. Motor function was preserved (MRC = 5) only in 22 (31%) patients while a severe impairment (MRC 1–3) was observed in 28 (29.4%) patients; sphincter dysfunction was found in 18 (25%) patients. More than a half of the patients (57.1%) presented with a reduced walking distance (<200 m) with 30.1% able to walk for <10 m.

### 3.2. Radiological Findings

Mean tumor volume was on average 25.8 ± 11.7 cm^3^, with the most common location being the thoracic spine in 62 (87.3%) cases. A multi-level tumor was reported in 31 (43.7%) patients and only 4 (5.6%) patients presented multiples SMs. Ventral and dorsal attachment were found in 29 (42.6%) and 28 (41.2%) patients respectively, while the majority did not present the pathognomonic sign of dural tail (74.6%). The tumors were characterized by intralesional calcifications in 14 (22.2%) cases. Preoperative T2-weighted images changes of the spinal cord were present in 23 (39.0%) patients.

### 3.3. Surgical and Pathological Data

Intraoperative monitoring was used in 25 (34.7%) patients. Complete tumor resection (Simpson I or II) was achieved in 67 (95.7%) cases. Intraoperative complications were reported in 7 (9.9%) patients, including 4 (5.6%) root lesions, 2 (2.8%) excessive bleeding, and 1 (1.4%) spinal cord injury. Postoperative complication was found in 12 (16.7%) patients, with the need for a reoperation in 5 (6.9%) patients. Among postoperative complications we had 3 (4.1%) infections/wound dehiscence, 1 (1.4%) pulmonary embolism, and 9 (12.5%) neurological deficits. The most common histological types were transitional and psammomatous meningioma (47.0% and 35.3% respectively), with 98.6% of the tumors being WHO grade 1.

### 3.4. Outcome Results

At last follow-up a good overall status (mod McCormick = I–II) was reported in 47 (65.3%) patients. A total of 43 (59.7%) patients showed a good relative outcome (Δ modified McCormick >0). In the vast majority of patient (86.1%) surgery had a positive impact, in terms of mod McCormick, either showing an improvement or maintaining a good preoperative status. T2-weighted image changes of the spinal cord on follow-up MRI were observed in 22 patients (44.9%). All descriptive data are shown in Table 2.

#### 3.4.1. Relative Outcome

To identify clinical variables associated with improved postoperative neurological status, we performed univariate statistical analysis after stratification between patients who showed a positive Δ mod McCormick (*n* = 43; improved group) and patients who had a Δ modified McCormick ≤0 (*n* = 29). Univariate analyses are entirely reported in Table 3. Sex, age, presence of comorbidities, ASA score ≥3, BMI, and previous surgery for spinal meningioma did not have any significant impact on postoperative functional improvement. Impaired preoperative motor function and high value (disability) of mod McCormick showed correlations with a postoperative functional improvement (*p* < 0.0001 both). Preoperative sphincter dysfunction was more likely seen in the improved group (37.2% vs. 6.9%, *p* = 0.0047). A shorter walking distance (≤200 m) before surgery was often observed in the group with good relative outcomes (77.8% vs. 29.6%, *p* = 0.0002). Neither tumor volume nor tumor location showed a significant difference between the groups. Multiple-level involvement was more represented in the improved group (53.5% vs. 28.6%, *p* = 0.0384). Postoperative neurological complications occurred less frequently in the improved group (4.7% vs. 24.1%, *p* = 0.0255). Overall, these data suggest that patients with worse preoperative neurological status are more likely to improve it after surgery for SM.

To test the independent association of the variables identified in univariate analyses with neurological outcome, we performed multivariate analyses. Here we included as covariates all the significant variables of the univariate analyzes, except the walking distance and postop MRI T2-weighted images changes of spinal cord due to the many missing data. Multivariate analysis showed that preoperative mod McCormick was the most significant independent predictor of improved outcome (coeff = 0.597, *p* < 0.0001). Patient’s age showed a statistically significant negative correlation with postoperative improvement (coefficient = −0.058, *p* < 0.0251). All the variables included in the multivariate model are reported in Table 4 and Figure 1.

#### 3.4.2. Overall Status

To identify clinical variables associated with good postoperative neurological status, we performed univariate statistical analysis between patients who showed a postoperative good overall status considered as modified McCormick = I–II (*n* = 47) and the group who had a poor overall status with a modified McCormick ≥ III (*n* = 25). Univariate analyses are entirely reported in Table 3. Median age was superior in the poor status group (82 vs. 77 years, *p* = 0.0007). Sex, presence of comorbidities, ASA score ≥ 3, BMI, duration of symptoms, sphincter dysfunction, and previous surgery for spinal meningioma did not have any significant impact on postoperative functional improvement. Preoperative good motor function (MRC = 4–5) was observed more frequently in the good outcome group (80.5% vs. 24.0%, *p* < 0.0001), as well as good preoperative modified McCormick (59.6% vs. 20.0%, *p* < 0.0001). A longer walking distance (>200 m) before surgery was often observed in the group with a good overall status (57.5% vs. 17.4%, *p* < 0.0001). Neither tumor volume nor tumor location showed a significant difference between the groups. Multiple-level involvement and postoperative neurological complications did not show differences between the two groups. Postoperative MRI T2WI hyperintensity were significantly less frequent in the good overall status group (32.4% vs. 73.3%, *p* = 0.0159).

Multivariate analysis showed that preoperative modified McCormick was the best independent predictor of good overall status (coeff = 0.402, *p* = 0.00027). Patient’s age showed a statistically significant correlation with good overall status (coeff = 0.058, *p* = 0.02507). All the variables included in the multivariate model are reported in Table 4 and Figure 2. In the multivariate analyses we included all the significant variables of the univariate analyses, except the walking distance and postoperative MRI T2-weighted image changes of the spinal cord, due to many missing data.

## 4. Discussion

### 4.1. Key Results

In this multicentric international retrospective study involving 72 elderly patients operated for resection of a SM, we found that (1) surgical resection had a good impact on patients’ function outcomes (86.1% either showing an improvement or maintaining a good preoperative status); (2) a good overall neurological status was achieved in the majority of patients (65.3%); (3) functional improvement (good relative outcome) occurred in 59.7% of the patients, while in the subgroup of patients who did not improve, 26.4% of them still showed a modified McCormick I or II; (4) preoperative mod McCormick was the best predictor for both outcome measures (coefficient = 0.597 relative outcome; coefficient = 0.402 overall status); (5) age demonstrated a correlation with both postoperative improvement (R = −0.058) and good overall status (coefficient = 0.058) but with a very low coefficient; (6) Finally, BMI, ASA score, comorbidities, duration of symptoms, tumor volume, localization, and histological diagnosis did not show any correlations with outcome.

### 4.2. Interpretation

Focusing on functional outcome is of great importance in surgical treatment of spinal meningioma in elderly. Aggressive spinal meningiomas are infrequent and recurrence is uncommon [29], but long-term disability in old patients is associated with low life expectancy [16]. Neurosurgeons might be reluctant to offer surgery to patients, expecting a more difficult recovery from symptoms and higher rate of postoperative complications undermining global outcome [8]. In high-income countries, elderly people tend to have more active lifestyles, longer lifespans, and higher levels of expected quality of life. However, the aging process is characterized by changes in the vital organs inducing the reduction or loss of functional reserve and compliance, which may become clinically manifest due to general anesthesia, operatory positioning, and surgery. Furthermore, the literature demonstrates that elderly patients underwent major elective surgery in many field (e.g., cardiac, vascular, orthopedic, oncological) without an increased rate of postoperative complications thanks to a dedicated preoperative anesthesiologic evaluation for risk assessments and intraoperative specific strategies [17,30,31].

Age limit to define a patient as elderly is not stated in literature and we decided to consider the most advanced age studied, including patients ≥ 75 years old, according to the series published by Sacko [10]. In our series, patient’s age showed a negative significant correlation with postoperative improvement (coefficient = −0.058, *p* < 0.0251, relative outcome), and a positive significant correlation with their functional status at follow-up (coefficient = 0.058, *p* < 0.02507, overall status). Projecting these values in our daily practice, it means that, after adjusting for the other covariates, approximately 17 years of age corresponds to one postoperative point of mod McCormick. For example, a patient who is 92 years old compared with one who is 75 will likely have 1 additional point of postoperative modified McCormick, and will be likely to improve 1 point less. In this view, older patients seem more fragile, and age is a factor which needs to be considered while considering surgery, but it seems not strong enough to constitute an absolute contraindication.

Otherwise, we did not find any correlation of the ASA score, BMI, or major comorbidities with postoperative outcome, according to previous studies [9,10]. Postoperative general complications (i.e., thrombosis, pulmonary embolism, pneumonia…) are rarely reported and seem not to impact patients’ outcomes as well as in previous series [8,26]. Analysis of a French database, which presents a large series of elderly patients, showed favorable outcomes notwithstanding level of comorbidities [11] and an absolute excess risk of mortality after spinal meningioma surgery [32].

Radiological findings were not specified in series focalized on the elderly. Recently, some studies [6,18,26,33,34,35,36,37,38,39,40], which include younger patients, analyzed MRI preoperative features as predictors of outcome. In particular, Baro showed that patients with larger tumors and higher spinal cord compression may present a lower preoperative functional status and may be prone to a worse clinical outcome; furthermore, preoperative T2 cord-signal changes are correlated with a poorer outcome [18]. Herein, in our cohort, we did not find correlations between preoperative MRI features and neurological outcomes, possibly due to the smaller sample and missing data. Conversely the postoperative T2WI hyper intensity showed correlations in univariate analysis, though not possible to confirm by multivariate analysis, also possibly due to missing data.

Preoperative modified McCormick appeared to be the most reliable predictor both for relative outcome (coefficient = 0.597, *p* < 0.0001) and overall status (coefficient = 0.402, *p* = 0.00027), Figure 1. This is explained by the fact that patients presenting with a high preoperative modified McCormick (III–IV) are less likely to obtain a good overall postoperative status (I–II), compared with patients with good preoperative statuses. Nonetheless, these patients are more likely to gain positive functional points (relative outcome) with surgery.

According to these results, surgery for elderly patients with poor functional status at presentation (modified McCormick IV–V) may be questionable because of their lower capacity to recover. Nevertheless, these patients have a potential of functional improvement which may ameliorate their quality of life. Accordingly, a though and careful patient-specific analysis taking into consideration patient’s frailty, social condition, and will to undergo surgical resection must be considered to obtain adequate preoperative counseling.

### 4.3. Generalizability

We reported and analyzed data from a cohort of homogenous elderly patients operated on for resection of spinal meningioma in four European referral centers, including 72 patients with only 2 potentially eligible patients excluded because they were lost in follow-up. This makes our series one of the largest reported in the literature. Even if included patients were operated over a period of 20 years, mainly due to the rarity of the pathology, both microsurgical techniques adopted and patient management were unchanged. We adopted the modified McCormick scale, which is widely used as a standard outcome tool in cases of patients affected by intradural spinal tumors [41].

### 4.4. Limitations

The retrospective nature of this study, together with the nonhomogeneous management strategies without random assignment, needs to be considered when evaluating the results.

We limited statistical bias by including missing data in the analysis as a factor.

## 5. Conclusions

According to the results from this multicentric study, we found that surgical resection of spinal meningiomas provided good outcomes in patients ≥75 years of age. Younger patients and patients with a better preoperative status showed better outcomes at follow-up. However, patients with worse preoperative statuses had greater relative outcomes after surgery. Advanced age and the presence of age-related comorbidities should not a priori contraindicate the surgical treatment, which could improve their quality of life.

## Figures and Tables

**Figure 1 cancers-14-04790-f001:**
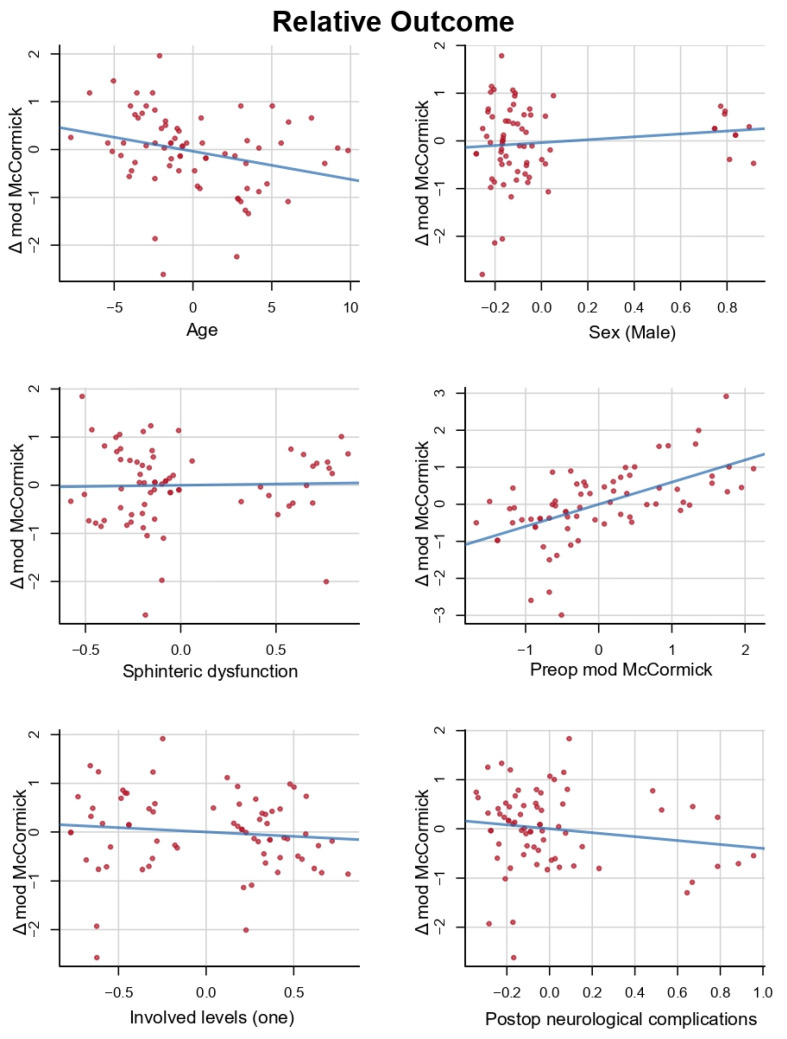
Variables included in the multivariate model for the relative outcome. Multivariate analysis showed that preoperative mod McCormick was the most significant independent predictor of improved outcome (Δ mod McCormick, coeff = 0.597, *p* < 0.0001). Patient’s age showed a statistically significant negative correlation with postoperative improvement (Δ mod McCormick, coeff = −0.058, *p* < 0.0251).

**Figure 2 cancers-14-04790-f002:**
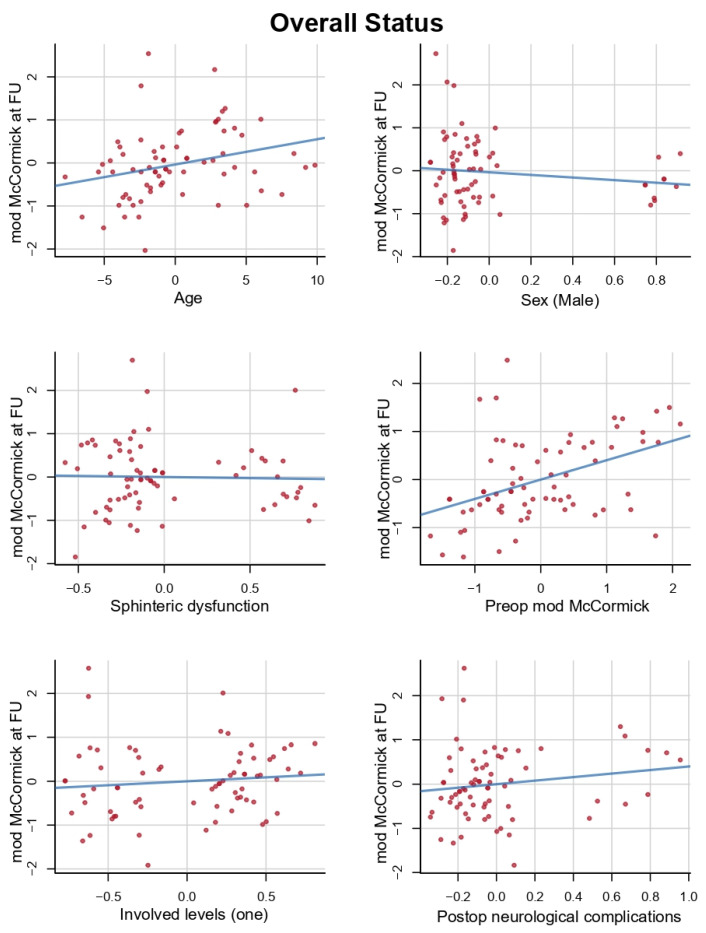
Variables included in the multivariate model for overall status. Multivariate analysis showed that preoperative modified McCormick was the best independent predictor of good overall status (coeff = 0.402, *p* = 0.00027). Patient’s age showed a statistically significant correlation with good overall status (coeff = 0.058, *p* = 0.02507).

**Table 1 cancers-14-04790-t001:** Modified McCormick Scale.

Grade	Clinical Characteristics
**I**	Neurologically intact, ambulates normally, may have minimal dysesthesia
**II**	Mild motor or sensory deficit; patient maintains functional independence
**III**	Moderate deficit, limitation of function, independent with external aid
**IV**	Severe motor or sensory deficit, limit of function with a dependent patient
**V**	Paraplegic or quadriplegic, even if there is flickering movement

**Table 2 cancers-14-04790-t002:** The cohort’s characteristics (*n* = 72).

	Overall
**Patients, *n* (%)**	72 (100)
**Age, median [Q1–Q3]**	78.5 [77.0–84.0]
**Sex, *n* (%)**	
** ** **Female**	61 (84.7)
** ** **Male**	11 (15.3)
**Duration of symptoms, median [Q1–Q3]**	6.0 [3.0–12.0]
** ** **<9 months, *n* (%) #**	44 (64.7)
** ** **≥9 months, *n* (%) #**	24 (35.3)
** ** **Missing data *n* (%) #**	4 (5.6)
**MRC scale, *n* (%) #**	
** ** **1**	9 (12.7)
** ** **2**	9 (12.7)
** ** **3**	10 (14.0)
** ** **4**	21 (29.6)
** ** **5**	22 (31.0)
** ** **Missing data**	1 (1.4)
**Sphincteric dysfunction, *n* (%)**	
** ** **No**	54 (75.0)
** ** **Yes**	18 (25.0)
**Preoperative Modified McCormick, *n* (%)**	
** ** **I**	17 (23.6)
** ** **II**	16 (22.2)
** ** **III**	22 (30.6)
** ** **IV**	11 (15.3)
** ** **V**	6 (8.3)
**Walking distance (wd), *n* (%) #**	
** ** **>200 m**	27 (42.9)
** ** **10 m ≤ wd ≤ 200 m**	17 (27.0)
** ** **<10 m**	19 (30.1)
** ** **Missing data**	9 (12.5)
**NF II, *n* (%)**	1 (1.4)
**Previous surgery, *n* (%)**	5 (6.9)
**Major comorbidities, *n* (%) #**	
** ** **No**	44 (62.0)
** ** **Yes**	27 (38.0)
** ** **Missing data**	1 (1.4)
**BMI, mean ± SD**	31.4 ±17.9
**ASA score, *n* (%) #**	
** ** **1–2**	27 (58.7)
** ** **3–4**	19 (41.3)
** ** **Missing data**	26 (36.1)
**Volume (cm^3^)**	
** ** **Mean ± SD**	25.8 ±11.7
**Localization, *n* (%) #**	
** ** **Cervical**	9 (12.7)
** ** **Dorsal**	62 (87.3)
** ** **Missing data**	1 (1.4)
**Involved Levels, *n* (%) #**	
** ** **1**	40 (56.3)
** ** **≥2**	31 (43.7)
** ** **Missing data**	1 (1.4)
**Dural attachment, *n* (%) #**	
** ** **Ventral**	29 (42.6)
** ** **Dorsal**	28 (41.2)
** ** **Lateral**	11 (16.2)
** ** **Missing data**	4 (5.6)
**Dural tails, *n* (%) #**	
** ** **Yes**	15 (25.4)
** ** **No**	44 (74.6)
** ** **Missing data**	13 (18.1)
**MRI T2 signal hyperintensity, *n* (%) #**	
** ** **Yes**	23 (39.0)
** ** **No**	36 (61.0)
** ** **Missing data**	13 (18.1)
**Calcifications, *n* (%) #**	
** ** **Yes**	14 (22.2)
** ** **No**	49 (77.8)
** ** **Missing data**	9 (12.5)
**Multiple meningiomas, *n* (%)**	4 (5.6)
**Simpson Grade, *n* (%) #**	
** ** **1**	16 (22.9)
** ** **≥2**	54 (77.1)
** ** **Missing data**	2 (2.8)
**WHO grade, *n* (%)**	
** ** **I**	71 (98.6)
** ** **II**	1 (1.4)
**Intraoperative Complications, *n* (%) #**	
** ** **Yes**	7 (9.9)
** ** **No**	64 (90.1)
** ** **Missing data**	1 (1.4)
**Intraoperative Neuromonitoring, *n* (%)**	25 (34.7)
**Histologic type, *n* (%) #**	
** ** **Angiomatous**	1 (1.5)
** ** **Fibroblastic**	4 (5.9)
** ** **Meningothelial**	3 (4.4)
** ** **Psammomatous**	24 (35.3)
** ** **Transitional**	32 (47.0)
** ** **Psammomatous + Transitional**	4 (5.9)
** ** **Missing data**	4 (5.6)
**Postoperative complications, *n* (%)**	12 (16.7)
**Surgery for complications, *n* (%)**	5 (6.9)
**Modified McCormick at last Follow up, *n* (%)**	
** ** **I**	36 (50.0)
** ** **II**	11 (15.3)
** ** **III**	21 (29.2)
** ** **IV**	4 (5.6)
** ** **V**	0 (0.0)
**Postoperative MRI T2 signal hyperintensity, *n* (%) #**	
** ** **Yes**	22 (44.9)
** ** **No**	27 (55.1)
** ** **Missing data**	23 (31.9)

# Percentage calculated excluding missing data. ASA: American Society of Anesthesiologists physical status classification system; BMI: Body Mass Index; IQR: interquartile range; MRC: Medical Research Council’s scale for muscles power; MRI: Magnetic resonance imaging, NF II: Neurofibromatosis type 2; SD: standard deviation; WHO grade: World health organization classification of tumors of the central nervous system.

**Table 3 cancers-14-04790-t003:** Univariate predictors of postoperative outcome improvement (represented by positive Δ modified McCormick) and overall status (estimated with follow-up modified McCormick) (*n* = 72).

	Relative Outcome			Overall Status		
	Not Improved ≤ 0	Improved > 0	*p*-Value	Poor ≥ III	Good = I–II	*p*-Value
**Patients, *n* (%)**	29 (40.3)	43 (59.7)		25 (34.7)	47 (65.3)	
**Age, median [Q1–Q3]**	78.0 [76.0–83.0]	79.0 [77.0–84.0]	0.2812	82.0 [81.0–85.0]	77.0 [76.0–82.0]	0.0007
**Sex, *n* (%)**			0.507			0.309
** Female **	26 (89.7)	35 (81.4)		23 (92.0)	38 (80.9)	
** Male**	3 (10.3)	8 (18.6)		2 (8.0)	9 (19.1)	
**Duration of symptoms, median [Q1–Q3]**	6.50 [2.0–11.5]	6.0 [3.3–11.8]	0.8658	7.5 [3.8–13.5]	6.0 [3.0–10.0]	0.3466
** <9 months, *n* (%) #**	16 (61.5)	28 (66.7)		12 (54.5)	32 (69.6)	
** ≥9 months, *n* (%) #**	10 (38.5)	14 (33.3)		10 (45.5)	14 (30.4)	
** Missing data, *n* (%) #**	3 (10.3)	1 (2.3)		1 (4.3)	3 (6.1)	
**MRC scale, *n* (%) #**			<0.0001			<0.0001
** 1**	0 (0.0)	9 (21.5)		9 (36.0)	0 (0.0)	
** 2**	1 (3.4)	8 (19.0)		4 (16.0)	5 (10.9)	
** 3**	4 (13.8)	6 (14.3)		6 (24.0)	4 (8.6)	
** 4**	6 (20.7)	15 (35.7)		4 (16.0)	17 (37.0)	
** 5**	18 (62.1)	4 (9.5)		2 (8.0)	20 (43.5)	
** Missing data**	0 (0.0)	1 (2.3)		0 (0.0)	1 (2.1)	
**Sphincteric disfunction, *n* (%)**			0.0047			0.3942
** Yes**	2 (6.9)	16 (37.2)		8 (32.0)	10 (21.3)	
** No**	27 (93.1)	27 (62.8)		17 (68.0)	37 (78.7)	
**Preoperative Modified McCormick, *n* (%)**			<0.0001			<0.0001
** I**	17 (58.6)	0 (0.0)		2 (8.0)	15 (31.9)	
** II**	7 (24.1)	9 (20.9)		3 (12.0)	13 (27.7)	
** III**	5 (17.2)	17 (39.5)		5 (20.0)	17 (36.2)	
** IV**	0 (0.0)	11 (25.6)		10 (40.0)	1 (2.1)	
** V**	0 (0.0)	6 (14.0)		5 (20.0)	1 (2.1)	
**Walking distance (wd) *n* (%) #**			0.0002			<0.0001
** >200 m**	19 (70.4)	8 (22.2)		4 (17.4)	23 (57.5)	
** 10 m ≤ wd ≤ 200 m**	6 (22.2)	11 (30.6)		4 (17.4)	13 (32.5)	
** <10 m**	2 (7.4)	17 (47.2)		15 (65.2)	4 (10.0)	
** Missing data**	2 (6.9)	7 (16.3)		2 (7.7)	7 (14.9)	
**Previous surgery, *n* (%)**			1			1
** Yes**	2 (6.9)	3 (7.0)		2 (8.0	3 (6.4)	
** No**	27 (93.1)	40 (93.0)		23 (92.0)	44 (93.6)	
**Major comorbidities, *n* (%) #**			0.7788			0.2923
** No**	19 (65.5)	25 (59.5)		12 (48.0)	15 (32.6)	
** Yes**	10 (34.5)	17 (40.5)		13 (52.0)	31 (67.4)	
** Missing data**	0 (0.0)	1 (2.3)		0 (0.0)	1 (2.1)	
**BMI (mean ± SD)**	32.57 ± (19.06)	30.49 ± (17.14)	0.641	35.57 ± (17.65)	29.16 ± (17.77)	0.1649
**ASA score, *n* (%) #**			0.1559			0.2863
** 1–2**	12 (42.9)	15 (35.7)		9 (37.5)	18 (39.1)	
** 3–4**	10 (35.7)	9 (21.4)		9 (37.5)	10 (21.7)	
** Missing data**	6 (21.4)	18 (42.9)		6 (25.0)	18 (39.1)	
**Volume (cm^3^) **						
** mean ± SD**	25.28 ± (12.51)	26.05 ± (11.17)	0.7852	25.52 ± (11.60)	25.85 ± (11.80)	0.9095
**Localization, *n* (%) #**			0.2752			0.1842
** Cervical**	2 (7.1)	7 (16.3)		1 (4.0)	8 (17.4)	
** Dorsal**	26 (92.9)	36 (83.7)		24 (96.0)	38 (82.6)	
** Missing data**	1 (3.4)	0 (0.0)		0 (0.0)	1 (2.1)	
**Involved Levels, *n* (%) #**			0.0384			0.7544
** 1**	20 (71.4)	20 (46.5)		13 (52.0)	27 (58.7)	
** ≥2**	8 (28.6)	23 (53.5)		12 (48.0)	19 (41.3)	
** Missing data**	1 (3.4)	0 (0.0)		0 (0.0)	1 (2.1)	
**Dural attachment, *n* (%) #**			0.3736			0.937
** Ventral**	14 (51.9)	15 (36.5)		10 (41.7)	19 (43.2)	
** Dorsal**	11 (40.7)	17 (41.5)		11 (45.8)	17 (38.6)	
** Lateral**	2 (7.4)	9 (22.0)		3 (12.5)	8 (18.2)	
** Missing data**	2 (6.9)	2 (4.6)		1 (4)	3 (6.4)	
**Dural tails, *n* (%) #**			0.5889			0.2082
** Yes**	6 (27.3)	9 (24.3)		7 (30.4)	8 (22.2)	
** No**	16 (72.7)	28 (75.7)		16 (69.6)	28 (77.8)	
** Missing data**	7 (–)	6 (–)		2 (–)	11 (–)	
**MRI T2 signal hyperintensity, *n* (%) #**			0.1939			0.135
** Yes**	6 (27.3)	17 (45.9)		11 (47.8)	12 (33.3)	
** No**	16 (72.7)	20 (54.1)		12 (52.2)	24 (66.7)	
** Missing data**	7 (24.1)	6 (13.9)		2 (8)	11 (23.4)	
**Calcifications, *n* (%) #**			0.8133			0.2108
** Yes**	5 (19.2)	9 (24.3)		4 (16.7)	10 (25.6)	
** No**	21 (80.8)	28 (75.7)		20 (83.3)	29 (74.4)	
** Missing data**	3 (10.3)	6 (16.2)		1 (4)	8 (17)	
**Multiple meningiomas, *n* (%)**	2 (6.8)	2 (4.6)	1	3 (12.0)	1 (2.1)	0.3164
**Simpson Grade, *n* (%) #**			0.891			1
** 1**	7 (25.0)	9 (21.4)		5 (20.8)	11 (23.9)	
** ≥2**	21 (75.0)	33 (78.6)		19 (79.2)	35 (76.1)	
** Missing data**	1 (3.4)	1 (2.3)		1 (4)	1 (2.1)	
**WHO grade, *n* (%)**			1			1
** 1**	29 (100.0)	42 (97.7)		25 (100.0)	46 (97.9)	
** 2**	0 (0.0)	1 (2.3)		0 (0.0)	1 (2.1)	
**Histologic type, *n* (%) #**			0.1795			0.6017
** Angiomatous**	0 (0.0)	1 (2.5)		0 (0.0)	1 (2.3)	
** Fibroblastic**	3 (10.7)	1 (2.5)		1 (4.2)	3 (6.8)	
** Meningothelial**	0 (0.0)	3 (7.5)		1 (4.2)	2 (4.5)	
** Psammomatous**	7 (25.0)	17 (42.5)		10 (41.7)	14 (31.8)	
** Transitional**	15 (53.6)	17 (42.5)		9 (37.5)	23 (52.3)	
** Transitional + Psammomatous**	3 (10.7)	1 (2.5)		3 (12.4)	1 (2.3)	
** Missing data**	1 (3.4)	3 (7)		1 (4)	3 (6.4)	
**Intraoperative Complications, *n* (%) #**			0.3061			0.3884
** Yes**	4 (14.3)	3 (7.0)		4 (16.0)	3 (6.5)	
** No**	24 (85.7)	40 (93.0)		21 (84.0)	43 (93.5)	
** Missing data**	1 (3.4)	0 (0.0)		0 (0.0)	1 (2.2)	
**Intraoperative Neuromonitoring, *n* (%)**	8 (27.6)	17 (39.5)	0.3046	13 (52.0)	12 (25.5)	0.0504
**Postoperative neurological complications, *n* (%)**	7 (24.1)	2 (4.7)	0.0255	3 (12.0)	6 (12.8)	1
**Surgery for complications, *n* (%)**	3 (10.3)	2 (4.7)	0.3861	2 (8.0)	3 (6.4)	1
**Modified McCormick at last Follow up, *n* (%)**			0.9423			<0.0001
** I**	14 (48.3)	22 (51.2)		0 (0.0)	36 (76.6)	
** II**	5 (17.2)	6 (14.0)		0 (0.0)	11 (23.4)	
** III**	8 (27.6)	13 (30.2)		21 (84.0)	0 (0.0)	
** IV**	2 (6.9)	2 (4.7)		4 (16.0)	0 (0.0)	
** V**	0 (0.0)	0 (0.0)		0 (0.0)	0 (0.0)	
**Postoperative MRI T2 signal hyperintensity, *n* (%) #**			0.244			0.0159
** Yes**	6 (35.3)	16 (50.0)		11 (73.3)	11 (32.4)	
** No**	11 (64.7)	16 (50.0)		4 (16.7)	23 (67.6)	
** Missing data**	12 (41.4)	11 (25.6)		10 (40)	13 (27.6)	

# Percentage calculated excluding missing data. ASA: American Society of Anesthesiologists physical status classification system; BMI: Body Mass Index; IQR: interquartile range; MRC: Medical Research Council’s scale for muscles power; MRI: Magnetic resonance imaging; SD: standard deviation; WHO grade: World health organization classification of tumors of the central nervous system.

**Table 4 cancers-14-04790-t004:** Multivariate predictors of postoperative relative outcome improvement (represented by positive Δ modified McCormick) and overall status (estimated with follow-up modified McCormick). Multivariate linear regression analysis (*n* = 72).

Parameters	Threshold	Relative Outcome				Overall Status			
		Coefficient	Standard Error	t-Value	*p*-Value	Coefficient	Standard Error	t-Value	*p*-Value
**Age**	-	−0.05869	0.02558	−2.294	0.0251	0.05869	0.02558	2.294	0.02507
**Sex**	Male	0.30381	0.28714	1.058	0.294	−0.30381	0.28714	−1.058	0.294
**Sphincteric dysfunction**	Yes	0.05132	0.23976	0.214	0.8312	−0.05132	0.23976	−0.214	0.8312
**Preoperative Modified McCormick**	-	0.59747	0.1044	5.723	<0.0001	0.40253	0.1044	3.856	0.00027
**Involved Levels**	1	−0.17957	0.2119	−0.847	0.3999	0.17957	0.2119	0.847	0.39992
**Postoperative neurological complications**	Yes	−0.40043	0.32362	−1.237	0.2205	0.40043	0.32362	1.237	0.22049

## Data Availability

The data presented in this study are available on request from the corresponding author.

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
