# Peer review of "Surgical Treatment of Spinal Meningiomas in the Elderly (≥75 Years): Which Factors Affect the Neurological Outcome? An International Multicentric Study of 72 Cases"

_cancers, 2022, doi:10.3390/cancers14194790_

Round 1

Reviewer 1 Report

Dear authors,
After a careful revision of “Surgical Treatment of Spinal Meningiomas in the Elderly (≥75 Years): Which Factors Affect the Neurological Outcome? An International Multicentric Study of 72 Cases”
please find below some considerations.
Your article is eloquent, methodic, and well written. This multicentric study is based on a relevant sample size that can provide standard and clear recommendations. So, in my opinion, this manuscript has a great potential of interest for readers and for future perspectives.

Author Response

Point 1:

Dear authors,

After a careful revision of “Surgical Treatment of Spinal Meningiomas in the Elderly (≥75 Years): Which Factors Affect the Neurological Outcome? An International Multicentric Study of 72 Cases”

please find below some considerations.

Your article is eloquent, methodic, and well written. This multicentric study is based on a relevant sample size that can provide standard and clear recommendations. So, in my opinion, this manuscript has a great potential of interest for readers and for future perspectives.

Response 1:

Dear Reviewer, thank you for your kind consideration.

Best regards

Gabriele Capo

Reviewer 2 Report

It was my pleasure to review this very well written article by Capo and colleagues.

They joined forces to get a large number of patients with spinal meningiomas, a quite rare disease.

I have only one critic point. The authors searched for factors influencing the relative outcome, according to their definition improvement of the McCormick scale after surgery in cpmparis with the scale preoperatively; and found out that a worse McCormick scale preoperatively indicates an improvement. The thing is that patients with a good McCormick scale (1) cannot improve anymore by definition on this scale. This can been seen in table 3. In the “not improved” group there were 17 patients with a score of 1 (58.6%), while in the improved group there weren’t any such patients.

I think the authors must rethink this issue. For the surgeon the main question would be: “should I operate a patient or not?”. Maybe the authors can give us a hint on, which patients would profit from surgery and which not. Should we operate on patients with McCormick scale of 1? Or not?

Minor issue:

In table 3: Preoperative Modified McCormick… the figure are not in the correct order

Page 12/16 first paragraph of the discussion: the author should omit the sentence: “Authors should discuss 265 the results and how they can be interpreted from the perspective of previous studies and 266 of the working hypotheses. The findings and their implications should be discussed in the 267 broadest context possible. Future research directions may also be highlighted“

In the literature: citations 6 and 19 are identical.

Author Response

Response to Reviewer 2 Comments

Point 1:

It was my pleasure to review this very well written article by Capo and colleagues.

They joined forces to get a large number of patients with spinal meningiomas, a quite rare disease.

I have only one critic point. The authors searched for factors influencing the relative outcome, according to their definition improvement of the McCormick scale after surgery in cpmparis with the scale preoperatively; and found out that a worse McCormick scale preoperatively indicates an improvement. The thing is that patients with a good McCormick scale (1) cannot improve anymore by definition on this scale. This can been seen in table 3. In the “not improved” group there were 17 patients with a score of 1 (58.6%), while in the improved group there weren’t any such patients.

I think the authors must rethink this issue. For the surgeon the main question would be: “should I operate a patient or not?”. Maybe the authors can give us a hint on, which patients would profit from surgery and which not. Should we operate on patients with McCormick scale of 1? Or not?

Response 1

Dear Reviewer, thank you for your kind review and comments. The points raised allow us to reflect on the interpretation of the results. I will try to answer to your questions as best I can, point-by-point.

During the choice of the methodology of the study, we actually considered this possible critical point. We agree that patients with McCormick I cannot improve further; at the same time, patients with McCormick V cannot get any worse. In our view, this does not mean that we can eliminate the two groups from the analysis of “relative outcome”. This would result in a bias in the selection of cases and in the evaluation of prognostic factors of surgery. Furthermore, patients with McCormick I cannot improve but they can get worse, and this eventuality must be observed. Despite the critical issues, we believe it is useful to verify the evolution of this parameter in our patients, given its diffusivity and practicality.

In addition, recognizing the weak point of the methodology, we decided to add the evaluation of the “overall status” which is not affected by this aspect, and allows us to say how many patients have an acceptable neurological status (McCormick I and II) at follow up.

I may include this comment in the manuscript (chapter limitation) if it would clarify the reading and interpretation of the results.

We believe that this analysis may add decision-making elements for surgical indication in the elderly. Obviously, we do not expect our results bring a high level of recommendation, but they can be the starting point for considering the surgical risk factors and to guide the evaluation of old patients with spinal meningioma.

 Point 2:

In table 3: Preoperative Modified McCormick… the figure are not in the correct order

Response 2

Do you mean that the results were not aligned (corresponding line of value and name of variable)? I checked the tables and aligned the values.

Point 3

Page 12/16 first paragraph of the discussion: the author should omit the sentence: “Authors should discuss 265 the results and how they can be interpreted from the perspective of previous studies and 266 of the working hypotheses. The findings and their implications should be discussed in the 267 broadest context possible. Future research directions may also be highlighted“

Response 3

We erased the sentence. It was a misprint of the template, sorry.

Point 4

In the literature: citations 6 and 19 are identical.

Response 4

Yes, they are identical, it is a mistake. I ask to the editor if during the editing phase it could be changed. With the new format of paper, I can’t change automatically all the citations if I erase one reference from the list. If it is not possible, I will do it manually. Thank you.

Best regards

Gabriele Capo

Reviewer 3 Report

The authors present a reletively large retrospective multicenter study on 72 elderly patients (>=75 years of age) with spinal meningeomas. The aim of the study was to identify predictive factors for postoperative neurological outcome.

Uni- and multivariate analyses were conducted on variuos clinical and radiological parameters, outcome measures were the change of the modified McCormick score (preop. - postop.) and the mod. McCormick (stratified in good and worse outcome) at last follow-up. 

The results show that age and preoperative MCCormick were independent predictors of neurological outcome at 12 ± 16.6 months.

The topic is interesting given the increasing number of elderly patients and the manuscript is well written. The strength of the study is its relatively large sample size and the well conducted statistical analysis.

Furthermore, since the complication rate is relatively high, it would be interesting to get more details on intra- and perioperative complications.

Did neuromonitoring reduce the rate of neurological complications?

Author Response

Response to Reviewer 3 Comments

Point 1

The authors present a reletively large retrospective multicenter study on 72 elderly patients (>=75 years of age) with spinal meningeomas. The aim of the study was to identify predictive factors for postoperative neurological outcome.

Uni- and multivariate analyses were conducted on variuos clinical and radiological parameters, outcome measures were the change of the modified McCormick score (preop. - postop.) and the mod. McCormick (stratified in good and worse outcome) at last follow-up. 

The results show that age and preoperative MCCormick were independent predictors of neurological outcome at 12 ± 16.6 months.

The topic is interesting given the increasing number of elderly patients and the manuscript is well written. The strength of the study is its relatively large sample size and the well conducted statistical analysis.

Furthermore, since the complication rate is relatively high, it would be interesting to get more details on intra- and perioperative complications.

Did neuromonitoring reduce the rate of neurological complications?

Response 1

Dear reviewer, thank you for your kind review and comments.

The rate of postoperative complication was relatively high (16.4%), but intraoperative were quite low (7 patients, 9.9%).

Among intraoperative complications we had 4 root lesions, 2 excessive bleeding and 1 spinal cord injury.

Among postoperative complications we had 3 (4.1%) infections/wound dehiscence, 1 (1.4%) pulmonary embolism and 9 (12.5%) neurological deficit.

The total of surgery made under neuromonitoring were 25 (34.7%).

Finally, we didn’t find statistical difference for complications with and w/o neuromonitoring.

We usually recommend use of intraoperative neuromonitoring in our practice. It can be useful and can make surgery safe, but we can’t affirm it with our results.

We added complications details in the manuscript in section Surgical and pathological data, lines 169-173.

Best regards

Gabriele Capo

Round 2

Reviewer 2 Report

The authors answered my concerns in adequatly